# No Parameters Left Behind: Sensitivity Guided Adaptive Learning Rate for Training Large Transformer Models

**Chen Liang\***, **Haoming Jiang\***[†], **Simiao Zuo\***, **Pengcheng He**[⋆], **Xiaodong Liu**[◇],
**Jianfeng Gao**[◇], **Weizhu Chen**[⋆] **& Tuo Zhao\***
\* Georgia Institute of Technology, [†] Amazon, [⋆] Microsoft Azure AI, [◇] Microsoft Research
{cliang73,jianghm,simiaozuo,tourzhao}@gatech.edu
{penhe,xiaodl,jfgao,wzchen}@microsoft.com

## ABSTRACT

Recent research has shown the existence of significant redundancy in large Transformer models. One can prune the redundant parameters without significantly sacrificing the generalization performance. However, we question whether the redundant parameters could have contributed more if they were properly trained. To answer this question, we propose a novel training strategy that encourages all parameters to be trained sufficiently. Specifically, we adaptively adjust the learning rate for each parameter according to its sensitivity, a robust gradient-based measure reflecting this parameter's contribution to the model performance. A parameter with low sensitivity is redundant, and we improve its fitting by increasing its learning rate. In contrast, a parameter with high sensitivity is well-trained, and we regularize it by decreasing its learning rate to prevent further overfitting. We conduct extensive experiments on natural language understanding, neural machine translation, and image classification to demonstrate the effectiveness of the proposed schedule. Analysis shows that the proposed schedule indeed reduces the redundancy and improves generalization performance.[1]

## 1 INTRODUCTION

Large-scale Transformer models have achieved remarkable success in various fields. Performance of these models scales with their number of parameters, which can be up to hundreds of millions, e.g., BERT (Devlin et al., 2018), DeBERTa (He et al., 2020), GPT-3 (Brown et al., 2020). Recent research, however, has shown the existence of significant redundancy in the Transformer models (Michel et al., 2019; Fan et al., 2019; Wang et al., 2019; Chen et al., 2020; Sanh et al., 2020). For example, Sanh et al. (2020) removes around 90% of the parameters, and the models exhibit only a marginal performance drop.

The existence of redundancy can hurt the model performance. Recent works have demonstrated that the removal of the redundant parameters can lead to better generalization performance, a phenomenon observed in both small-scale models (Mozer & Smolensky, 1989; Rasmussen & Ghahramani, 2001; Grünwald & Grunwald, 2007) and large-scale Transformer models (Bartoldson et al., 2019; Voita et al., 2019; Hou et al., 2020; Liang et al., 2021). As illustrated in Figure 1, with up to 20% of the parameters pruned, the generalization performance boosts up to 1%.

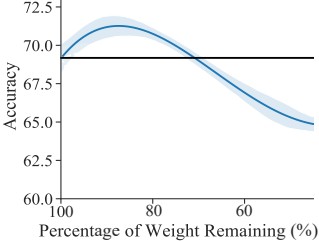

Figure 1: Validation results of fine-tuning BERT-base at different sparsity levels on the RTE dataset (Wang et al., 2018) in Liang et al. (2021). Solid black curve represents the full model performance.

As a result, we aim to improve model generalization through redundancy elimination. However, the existence of redundancy has long been regarded as inevitable. The common belief is that, in each network, there always exists a set of parameters "born" to be useless (Frankle & Carbin, 2018; Liu et al., 2018). Following this belief, pruning, where redundant parameters are directly zeroed out, becomes one of the most widely adopted solutions to redundancy elimination. However, we ask a critical question here:

---

[1]Our code has been released at https://github.com/cliang1453/SAGE

> *Are these parameters really redundant, or just insufficiently trained by commonly used training strategies?*

Our question is motivated by empirical observations, which show that training strategies indeed play a role in causing redundancy. For example, different learning rates (Table 1), random seeds and optimizers (Morcos et al., 2019) can produce models with similar performance but different sets of redundant parameters. This suggests that the redundancy of parameters depends on the training strategy: A training strategy often prefers specific parameters and provides them with sufficient training. In contrast, the other parameters receive insufficient training and become under-fitted. As a result, these parameters become redundant, such that they fail to contribute to the generalization and prevent the model from achieving its ideal performance. Therefore, we hypothesize that with a desirable training strategy, these redundant parameters can receive more sufficient training and become useful ultimately.

We verify the hypothesis by proposing a novel training strategy, which encourages all parameters to be trained sufficiently. Throughout the training process, we simultaneously excite the under-fitted parameters to reduce redundancy and regularize the well-fitted parameters to prevent overfitting.

| OVLP Among | Avg % OVLP |
|---|---|
| 2 Models | 59.8% |
| 3 Models | 46.5% |
| 5 Models | 35.7% |

Table 1: Percentage of overlapping between the $30\%$ most redundant parameters in 5 BERT-base models fine-tuned using $\{1, 5, 8, 10, 20\} \times 10^{-5}$ as learning rates on SST-2.

More specifically, we propose an adaptive learning rate schedule – SAGE (Sensitivity-guided Adaptive learninG ratE), where each parameter learns at its own pace guided by its sensitivity. Sensitivity originated in model pruning, where it is used to measure the redundancy of the parameters (Molchanov et al., 2016; 2019; Theis et al., 2018; Lee et al., 2018; Ding et al., 2019). In pruning literature, parameters with low sensitivity are considered redundant. Since a redundant parameter could be insufficiently trained and under-fitted, we promote its training by increasing its learning rate. In contrast, for a parameter with high sensitivity, i.e., it is considered sufficiently trained and well-fitted, we slow down its training by decreasing its learning rate to prevent overfitting.

Moreover, we introduce a local temporal variation of the sensitivity as a second factor to further guide the learning rate. The local temporal variation essentially measures the uncertainty of sensitivity, which mainly comes from two sources: (1) The sensitivity can have large variance due to data sampling. This is because during training, the sensitivity is evaluated using a randomly sampled mini-batch instead of all the training data. (2) The sensitivity of a parameter may not be stable and can vary drastically among iterations, which introduces extra uncertainty. We define the local temporal variation of a parameter as the absolute difference between its sensitivity and an exponential moving average of its sensitivity from all previous iterations. A large local temporal variation implies high uncertainty in the sensitivity at the current iteration, and therefore it is not yet a reliable indicator of redundancy. Accordingly, we should avoid significantly decreasing its learning rate even though its sensitivity at the current iteration might be large.

Therefore, we eventually require the overall learning rate schedule for each parameter to be proportional to the ratio between the local temporal variation and the sensitivity. This can effectively account for the uncertainty issue in sensitivity.

We conduct experiments on a wide range of tasks and models to demonstrate the effectiveness of SAGE. In natural language understanding, the fine-tuning performance of BERT-base (Devlin et al., 2018) and RoBERTa-large (Liu et al., 2019b) improves $1.4$ and $0.6$ task-average score on the dev set of the GLUE benchmark (Wang et al., 2018), respectively. Furthermore, SAGE improves neural machine translation performance using Transformer-base (Vaswani et al., 2017) on two datasets, suggesting it also benefits training-from-scratch. SAGE also boost the image classification accuracy on ImageNet dataset (Deng et al., 2009) with Vision Transformer models (Dosovitskiy et al., 2020). Furthermore, our experiments demonstrate SAGE is complementary to various types of optimizers, e.g., SGD (Robbins & Monro, 1951), Adam, and Adamax (Kingma & Ba, 2014).

Moreover, we observe several favorable proprieties of SAGE. First, it leads to balanced and sufficient training on all parameters and produces a better-generalized model. Second, SAGE is complementary to state-of-the-art training methods. Specifically, we show that SAGE achieves better performance on GLUE when combined with adversarial regularization (Jiang et al., 2019).

## 2 PRELIMINARY

We briefly review the sensitivity of the parameters and adaptive learning rate methods.

### 2.1 SENSITIVITY OF THE PARAMETERS

The sensitivity of a parameter essentially approximates the change in the loss magnitude when this parameter is completely zeroed-out (LeCun et al., 1990; Mozer & Smolensky, 1989). If the removal of a parameter causes a large influence on the loss, then the model is sensitive to it. More specifically, we define a deep neural network with parameters $\boldsymbol{\Theta} = [\theta_1, ..., \theta_J] \in \mathbb{R}^J$, where for $j = 1, ..., J$, $\theta_j \in \mathbb{R}$ denotes each parameter. We further define $\boldsymbol{\Theta}_{j,-j} = [0, ..., 0, \theta_j, 0, ..., 0] \in \mathbb{R}^J$. We denote the loss of the model as $L(\boldsymbol{\Theta})$, and the gradients of the loss with respect to $\boldsymbol{\Theta}$ as $\nabla_{\boldsymbol{\Theta}} L(\boldsymbol{\Theta})$. The sensitivity of the $j$-th parameter is defined as the magnitude of the gradient-weight product:

$$I_j = |\boldsymbol{\Theta}_{j,-j}^{\top} \nabla_{\boldsymbol{\Theta}} L(\boldsymbol{\Theta})|. \tag{1}$$

This definition is derived from the first-order Taylor expansion of $L(\cdot)$ with respect to $\theta_j$ at $\boldsymbol{\Theta}$. Specifically, $I_j$ approximates the absolute change of the loss given the removal of $\theta_j$:

$$\boldsymbol{\Theta}_{j,-j}^{\top} \nabla_{\boldsymbol{\Theta}} L(\boldsymbol{\Theta}) \approx L(\boldsymbol{\Theta}) - L(\boldsymbol{\Theta} - \boldsymbol{\Theta}_{j,-j}).$$

The sensitivity was originally introduced for model pruning (Molchanov et al., 2016; 2019; Theis et al., 2018; Lee et al., 2018; Ding et al., 2019; Xiao et al., 2019), and it was commonly used as an "importance score" for model weights. The parameters with high sensitivity are of high importance and should be kept (Lubana & Dick, 2020). Parameters with low sensitivity are considered redundant, and they can be safely pruned with only marginal influence on the model loss.

### 2.2 ADAPTIVE LEARNING RATE METHODS

Adaptive learning rate methods adjust the learning rate of each individual parameter based on the training progress. Most of these methods focus on adapting the training to the optimization landscape, e.g., AdaGrad (Duchi et al., 2011), AdaDelta (Zeiler, 2012), RMSProp (Hinton et al., 2012), Adam(Kingma & Ba, 2014) and RAdam (Liu et al., 2019a). Their purpose is to make the model converge faster to the first-order stationary solutions. Specifically, these methods prefer updating the weights with smaller second-order moments, as the loss function is generally flat along directions corresponding to such weights.

There are also some adaptive learning rate methods focusing on the perspective of improving model generalization (Loshchilov & Hutter, 2018; Foret et al., 2020). For example, AdamW (Loshchilov & Hutter, 2018) propose to decouple the weight decay and gradient update to avoid regularizing weights that have larger gradient magnitudes with a weaker strength.

## 3 METHOD

We introduce our proposed adaptive learning rate schedule, SAGE. Our method customizes a specific learning rate for each parameter at each iteration. A parameter's learning rate at a certain iteration is determined by two factors: sensitivity and its local temporal variation.

**Sensitivity of the parameters.** At the $t$-th iteration, following Eq. (1), we define the sensitivity of $\theta_j^{(t)}$ as

$$I_j^{(t)} = |\boldsymbol{\Theta}_{j,-j}^{(t)\top} \nabla_{\boldsymbol{\Theta}^{(t)}} L(\boldsymbol{\Theta}^{(t)})|, \tag{2}$$

which reflects the influence of removing $\theta_j^{(t)}$ in the model loss. In previous literature, $\theta_j^{(t)}$ is considered redundant when $I_j^{(t)}$ is small. In contrast, we hypothesize that $\theta_j^{(t)}$ is just insufficiently trained and under-fitted, and can become less redundant when receiving further training.

**Local temporal variation.** Recall that the sensitivity measure involves excessive uncertainty, which comes from: (1) Sensitivity is measured based on a randomly sampled mini-batch of the training data at each iteration, which leads to a large variance; (2) Sensitivity can be unstable and vary drastically, as changes of the model introduce extra uncertainty to the measure.

One way to measure the uncertainty of sensitivity of $\theta_j$ is the absolute change of sensitivity, i.e., $|I_j^{(t)} - I_j^{(t-1)}|$. Such a quantity often has a large variance in practice. Therefore, we propose to keep

track of an exponential moving average of $I_j^{(t)}$ as

$$\widehat{I}_j^{(t)} = \beta_0 \widehat{I}_j^{(t-1)} + (1 - \beta_0) I_j^{(t)},$$

where $\widehat{I}_j^{(0)} = 0$ and $\beta_0 \in (0, 1)$ is a hyper-parameter. Based on $\widehat{I}_j^{(t)}$, we measure the uncertainty of the $j$-th parameter's sensitivity using the local temporal variation defined as:

$$U_j^{(t)} = |I_j^{(t)} - \widehat{I}_j^{(t)}|. \tag{3}$$

We remark that a large $U_j^{(t)}$ implies that there exists high uncertainty in $I_j^{(t)}$, and therefore it is not yet a reliable indicator of the redundancy of $\theta_j^{(t)}$.

**Algorithm.** We denote the learning rate at the $t$-th iteration as $\eta^{(t)}$ under the original schedule. Then the sensitivity-guided learning rate for the $j$-th parameter at the $t$-th iteration can be computed as

$$\eta_j^{(t)} = \eta^{(t)} \cdot \frac{U_j^{(t)} + \epsilon}{\widehat{I}_j^{(t)} + \epsilon} = \eta^{(t)} \cdot \frac{|I_j^{(t)} - \widehat{I}_j^{(t)}| + \epsilon}{\widehat{I}_j^{(t)} + \epsilon}, \tag{4}$$

where $0 < \epsilon \ll 1$ prevents zero learning rate and zero denominator. Algorithm 2 shows the SAGE algorithm for SGD, and extensions to other algorithms, such as Adam (Kingma & Ba, 2014), are straightforward (Appendix A.4.1).

In Eq. (4), we place $\widehat{I}_j^{(t)}$ in the denominator, as one of our goals is to encourage all parameters to be sufficiently trained. If $\widehat{I}_j^{(t)}$ is small, we promote its training by increasing its learning rate. If $\widehat{I}_j^{(t)}$ is large, we slow down its training to prevent overfitting by decreasing its learning rate.

We place $U_j^{(t)}$ in the numerator to measure the uncertainty in the sensitivity. A large $U_j^{(t)}$ implies $I_j^{(t)}$ is not yet a reliable indicator of the redundancy in $\theta_j^{(t)}$. We thus avoid significantly decreasing its learning rate.

---

**Algorithm 1** SGD-SAGE ($\odot$ denotes Hadamard product and $\oslash$ denotes Hadamard division)

---

**Input:** Model parameters $\boldsymbol{\Theta} \in \mathbb{R}^J$; Data $\mathcal{D}$; Learning rate schedule $\eta(\cdot)$; Total training iteration $T$; Moving average coefficient $\beta_0$.
1: Initialize $\widehat{I}^{(0)} = \mathbf{0} \in \mathbb{R}^J$.
2: **for** $t = 1, ..., T$ **do**
3:     Sample a minibatch $b^{(t)}$ from $\mathcal{D}$.
4:     Compute gradient $\nabla_{\boldsymbol{\Theta}^{(t)}} L(b^{(t)}, \boldsymbol{\Theta}^{(t)})$.
5:     $I^{(t)} = |\boldsymbol{\Theta}^{(t)} \odot \nabla_{\boldsymbol{\Theta}^{(t)}} L(b^{(t)}, \boldsymbol{\Theta}^{(t)})|$.
6:     $\widehat{I}^{(t)} = \beta_0 \widehat{I}^{(t-1)} + (1 - \beta_0) I^{(t)}$.
7:     $U^{(t)} = |I^{(t)} - \widehat{I}^{(t)}|$.
8:     $\boldsymbol{\Theta}^{(t+1)} = \boldsymbol{\Theta}^{(t)} - \eta^{(t)} (U^{(t)} + \epsilon) \oslash (\widehat{I}^{(t)} + \epsilon) \odot \nabla_{\boldsymbol{\Theta}^{(t)}} L(b^{(t)}, \boldsymbol{\Theta}^{(t)})$.
9: **end for**

---

**Computation and memory usage.** SAGE adds a marginal cost to computation and memory usage. At each iteration, we only perform an extra element-wise multiplication between the weight matrix and the corresponding gradient matrix obtained through back-propagation. The only memory cost is to store the exponential moving average of sensitivity.

## 4 EXPERIMENTS

We evaluate SAGE on widely used benchmarks for natural language understanding (NLU), neural machine translation (NMT), and image classification.

### 4.1 NATURAL LANGUAGE UNDERSTANDING

**Model and data.** We evaluate the fine-tuning performance of the pre-trained language models, BERT-base (Devlin et al., 2018) and RoBERTa-large (Liu et al., 2019b), on the General Language Understanding Evaluation (GLUE, Wang et al. (2018)) benchmark. GLUE contains nine NLU tasks, including textual entailment, question answering, sentiment analysis, and text similarity. Details about the benchmark are deferred to Appendix A.1.1.

**Implementation Details.** We implement our method using the MT-DNN code-base[2]. We follow the suggested training and hyper-parameters settings from Liu et al. (2020). Specifically, we adopt Adam and Adamax (Kingma & Ba, 2014) with corrected weight decay (Loshchilov & Hutter, 2018) as the baseline optimizer and we set $\beta = (0.9, 0.999)$. We use a linear-decay learning rate schedule, and we apply SAGE to both Adam and Adamax.

We select learning rates in range of $\{1, 2, 3, 5, 8\} \times \{10^{-5}, 10^{-4}\}$. We select $\beta_0$ in range of $[0.6, 0.9]$ with an increment of $0.05$. Other training details are reported in Appendix A.1.2.

**Main results.** Table 2 and Table 3 show the evaluation results on the GLUE benchmark. The dev results are averaged over 5 different random seeds, and all gains are statistically significant[3]. We select the best single task model for test evaluation.

| Model | Optimizer | RTE Acc | MRPC Acc/F1 | CoLA Mcc | SST-2 Acc | STS-B P/S Corr | QNLI Acc | QQP Acc/F1 | MNLI-m/mm Acc | Average Score |
|---|---|---|---|---|---|---|---|---|---|---|
| BERT$_{\text{BASE}}$ | Devlin et al. (2018) | - | -/86.7 | - | 92.7 | -/- | 88.4 | -/- | 84.4/- | - |
| | Adam | 63.5 | 84.1/89.0 | 54.7 | 92.9 | 89.2/88.8 | 91.1 | 90.9/88.1 | 84.5/84.4 | 81.5 |
| | Adam-SAGE | **73.3** | **87.0/90.9** | **60.3** | **93.5** | **90.3/89.9** | **91.7** | **91.2/88.1** | **84.7/84.8** | **84.0** |
| | Adamax | 69.2 | 86.2/90.4 | 57.8 | 92.9 | 89.7/89.2 | 91.2 | 90.9/88.0 | 84.5/84.4 | 82.8 |
| | Adamax-SAGE | **74.0** | **87.3/91.0** | **59.7** | **93.8** | **90.3/89.8** | **91.8** | **91.2/88.2** | **85.0/85.2** | **84.2** |
| RoBERTa$_{\text{LARGE}}$ | Liu et al. (2019b) | 86.6 | -/90.9 | 68.0 | 96.4 | 92.4/- | 94.7 | 92.2/- | 90.2/90.2 | - |
| | Adamax | 86.6 | 90.4/93.1 | 67.5 | 96.4 | 92.4/92.2 | 94.7 | 92.1/89.3 | 90.4/90.3 | 88.7 |
| | Adamax-SAGE | **87.8** | **91.5/93.9** | **68.7** | **96.7** | **92.7/92.4** | **94.9** | **92.2/89.4** | **90.8/90.4** | **89.3** |

Table 2: Single task fine-tuning dev results on GLUE. All results are from our implementations. '-' denotes missing results.

Our method gains $1.4$ on dev and $1.1$ on test of the task-average score on BERT-base. In large datasets, i.e., MNLI (392K) and QNLI (108K), SAGE improves around $0.5$ points. In small datasets, i.e., RTE (2.5K) and CoLA (8.5K), we obtain more than 2 points of improvements. Such observations indicate that SAGE is very effective on the small datasets. Furthermore, SAGE improves upon RoBERTa-large by $0.6$ average scores, suggesting SAGE can still achieve significant improvements for larger and more adequately pre-trained models than BERT-base.

| | RTE Acc | MRPC F1 | CoLA Mcc | SST-2 Acc | STS-B P/S Corr | QNLI Acc | QQP F1 | MNLI-m/mm Acc | Average Score |
|---|---|---|---|---|---|---|---|---|---|
| BERT$_{\text{BASE}}$ (Devlin et al., 2018) | 66.4 | 88.9 | 52.1 | 93.5 | 85.8 | 90.5 | 71.2 | 84.6/83.4 | 79.6 |
| BERT$_{\text{BASE}}$, Adamax | 66.8 | 88.6 | 54.0 | 93.4 | 86.6 | 90.6 | 71.1 | 84.7/83.6 | 79.9 |
| BERT$_{\text{BASE}}$, Adamax-SAGE | **69.8** | **89.7** | **54.5** | **94.1** | **87.1** | **90.8** | **71.3** | **84.9/83.8** | **80.7** |

Table 3: Single task fine-tuning test results from the GLUE evaluation server.

| Model | Optimizer | IWSLT'14 De-En | WMT'16 En-De |
|---|---|---|---|
| Transformer$_{\text{BASE}}$ | Adam | 34.5 | 27.3 |
| | Adam-SAGE | **35.1** | **27.7** |

Table 4: Neural machine translation BLEU scores on test set. All results are from our implementation.

| Model | Optimizer | CIFAR100 | ImageNet |
|---|---|---|---|
| ViT-B/32 | SGD* | 91.97 | 81.28 |
| | SGD-SAGE | **92.68** | **81.72** |
| ViT-L/32 | SGD* | 93.04 | 80.99 |
| | SGD-SAGE | **93.74** | **81.90** |

Table 5: Image classification test accuracy. Results with $*$ are from Dosovitskiy et al. (2020). ViT-B/32 and ViT-L/32 each denotes ViT-base and ViT-large model with $32 \times 32$ input patch size.

---

[2]https://github.com/namisan/mt-dnn

[3]The dev results on RoBERTa-large are averaged over 3 different random seeds. All results have passed a paired student t-test with p-values less than 0.05. The detailed statistics are summarized in Appendix A.1.3.

## 4.2 Neural Machine Translation

**Model and Data.** We evaluate SAGE on the Transformer-base NMT models (Vaswani et al., 2017) using two widely used NMT datasets, IWSLT'14 De-En (Cettolo et al., 2015)[4] and WMT'16 En-De (Bojar et al., 2016)[5]. IWSLT'14 De-En is a low-resource dataset, which contains 160K sentence pairs. WMT'16 En-De is a rich-resource dataset, which contains 4.5M sentence pairs. Dataset and pre-processing details are deferred to Appendix A.2.1.

**Implementation Details.** We implement the algorithms using the *fairseq* code-base and follow the training and hyper-parameters settings from Ott et al. (2018; 2019). Specifically, we adopt the inverse square root learning rate schedule and we employ Adam (Kingma & Ba, 2014) as the optimizer with $\beta = (0.9, 0.98)$. We apply SAGE to the same setting.

We select learning rates in range of $\{5, 7\} \times 10^{-5} \cup \{1, 2\} \times 10^{-4}$ and select $\beta_0$ in range of $\{0.5, 0.6, 0.7, 0.8, 0.9\}$. Comprehensive training details are reported in Appendix A.2.2.

**Main results.** Table 4 shows the BLEU scores on the IWSLT'14 De-En and the WMT'16 En-De test set, where SAGE improves around $0.6$ and $0.4$ points, respectively. This suggests that other than fine-tuning, SAGE can also improve the generalization of trained-from-scratch models in both low-resource and rich-resource settings.

## 4.3 Image Classification

**Model and data.** We evaluate SAGE using Vision Transformer models (ViT) on the CIFAR100 (Krizhevsky et al., 2009) and ILSVRC-2012 ImageNet dataset (Deng et al., 2009). Specifically, we evaluate the fine-tuning performance of the ViT-base and ViT-large pre-trained using ImageNet-21k, a superset of ImageNet dataset with 21k classes and 14M images. Data and pre-processing details are deferred to Appendix A.3.1.

**Implementation details.** All experiments follow the suggested training configuration of Dosovitskiy et al. (2020) and a jax-implemented code base [6]. We adopt SGD as the baseline optimizer with a momentum factor 0.9. We fine-tune the models for 100K steps for CIFAR100, and 200K steps for ImageNet. We select learning rates in range of $\{0.02, 0.05, 0.08, 0.1\}$ and select $\beta_0$ in range of $\{0.85, 0.90, 0.95\}$. Comprehensive training details are reported in Appendix A.3.2.

**Main results.** Table 5 shows the evaluation results on CIFAR100 and ImageNet. SAGE outperforms baselines by a significant margin. This demonstrates that SAGE is quite general, and can be applied to various tasks (e.g., NLP and computer vision) and optimizers (e.g., Adam, Adamax and SGD).

## 5 Analysis

We verify that SAGE leads to more sufficient training (Section 5.1), better generalization performance (Section 5.2), and is complementary to existing state-of-the-art regularization methods (Section 5.3). We also provide ablation studies in Appendix A.4.4.

## 5.1 SAGE Leads to More Sufficient Training

Recall that SAGE adjusts the learning rate for each parameter according to two factors: the sensitivity of parameters and the local temporal variation of sensitivity. By inspecting these factors, we verify that SAGE leads to more sufficient training.

**The sensitivity distribution is more concentrated.** Figure 2 shows the sensitivity distribution of parameters in the SAGE optimized models and the baseline models. We select the hyper-parameters that yield the best generalization performance on the BERT-base model, and we evaluate the sensitivity of each parameter using the entire training set. See Appendix A.4.2 for implementation details.

We observe that the sensitivity distribution exhibits a lower variance in the SAGE optimized models than the baseline models. This suggests that the sensitivity of parameters becomes more concentrated. In other words, the amount of each parameter's contribution is more balanced, and the model is more sufficiently trained.

---

[4]https://wit3.fbk.eu/

[5]http://data.statmt.org/wmt16/translation-task/

[6]https://github.com/google-research/vision_transformer

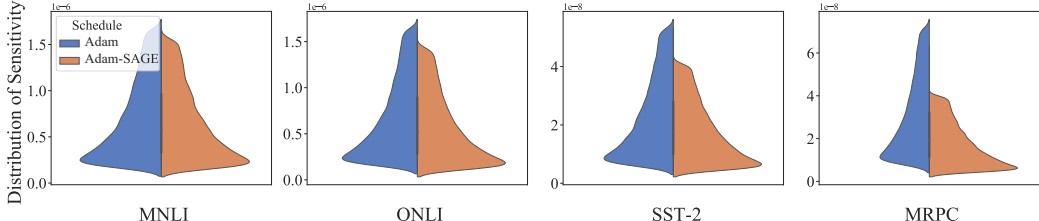

Figure 2: The sensitivity distribution of the BERT-base models fine-tuned on GLUE tasks. Note that we drop some outliers to ease visualization.

**Even the most redundant parameters contribute to the model performance.** Recall that sensitivity is a type of importance score in pruning, which is a straightforward approach to measure each parameter's contribution. Therefore, we conduct an unstructured, one-shot pruning experiment on the fine-tuned BERT-base models. Specifically, we remove up to $40\%$ parameters[7] with the lowest sensitivity scores and evaluate the pruned models' performance. We average the results over 5 models trained with different random seeds. Figure 3 *Upper* shows the generalization performance of the pruned models. To ease the comparison, Figure 3 *Lower* shows the change in generalization performance with respect to the un-pruned models.

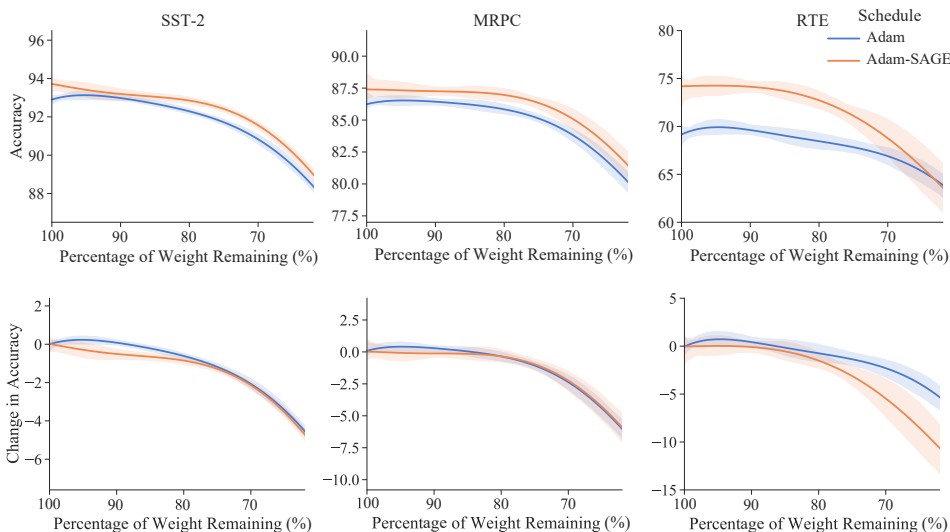

Figure 3: *Upper*: Model generalization performance at different pruning ratios; *Lower*: Change in generalization performance with respect to the full model. Pruning is conducted on the fine-tuned BERT-base models.

We have the following observations:

• The pruning performance of the SAGE optimized models remains higher than that of the baseline models (Figure 3 *Upper*).

• Even the most redundant parameters in the SAGE optimized models makes contributions (Figure 3 *Lower*). When there are over $80\%$ of weights remaining, the pruning performance of the baseline models is comparable or even superior than their un-pruned alternatives. In contrast, the performance of the SAGE optimized models consistently deteriorates. This suggests that the most redundant parameters in the baseline models fail to contribute, while those in the SAGE optimized models are trained more sufficiently and are able to make contributions.

**Sensitivity is a reliable indicator of redundancy.** We visualize the local temporal variation (Figure 4) to verify that sensitivity indeed becomes a more reliable indicator of redundancy in SAGE than in the baselines. We track the variation for all parameters in the BERT-base model at each iteration, and

---

[7]Embedding weights are excluded.

we evaluate the variation based on the current mini-batch of training data. See Appendix A.4.2 for implementation details.

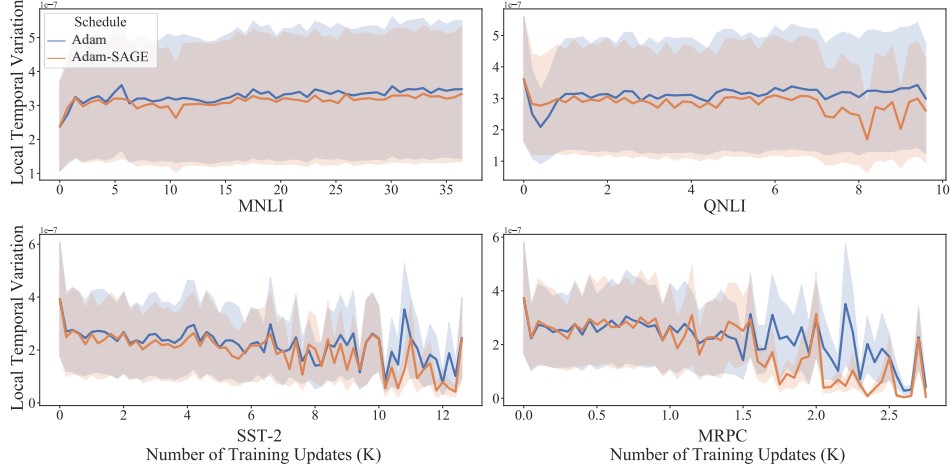

Figure 4: The local temporal variation of sensitivity (with $\beta_0 = 0.7$) during training.

We observe that the local temporal variation in SAGE remains lower or decreases faster than in the baselines for all tasks. For example, the variation in the baseline approach remains large in QNLI. In contrast, the variation in SAGE decreases, suggesting the sensitivity indeed stabilizes and becomes a reliable indicator of redundancy.

## 5.2 SAGE LEADS TO BETTER GENERALIZATION PERFORMANCE

We verify that SAGE leads to better generalization performance through inspecting the learning curves, decision boundary and hyper-parameter search space.

**Learning Curves.** Figure 6 shows the training loss, validation loss, learning rate, and sensitivity score obtained by fine-tuning BERT-base on SST-2. All experiment details are deferred to Appendix A.4.3. We have two major observations: 1) SAGE's validation loss descends faster and SAGE is less prone to overfitting. This observation suggests that SAGE has a regularization effect and reduces the model variance. 2) SAGE's variance of the sensitivity score becomes lower through training, aligning with our observation in Figure 2. This suggests that SAGE gives rise to a more balanced and sufficient training. Both observations agree with our initial motivation (Figure 1) that redundancy elimination can lead to better generalization.

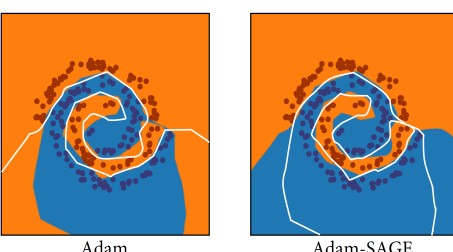

Figure 5: Decision boundary predicted on the Spiral dataset. The white curve on Adam-SAGE corresponds the decision boundary of Adam.

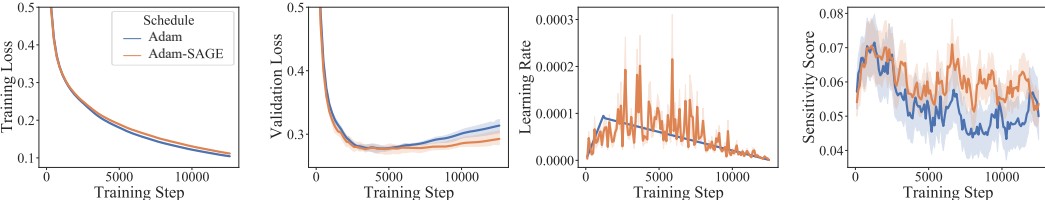

Figure 6: Learning curves obtained by fine-tuning BERT-base on SST-2 dataset.

**Hyper-parameter Study.** Figure 7 shows the validation accuracy heatmap obtained by fine-tuning BERT-base on the RTE dataset. We plot the accuracy obtained by training with different learning rates, Adam's $\beta$s and SAGE's $\beta_0$s. We can observe that SAGE consistently achieves a better generalization performance within a larger region of hyper-parameter search space under different $\beta_0$s. We also provide a hyper-parameter study for more datasets in Appendix A.4.5.

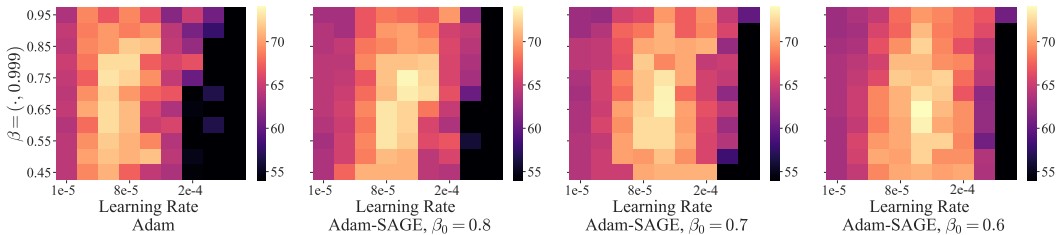

Figure 7: Validation accuracy obtained by fine-tuning BERT-base on RTE dataset with a wide range of hyper-parameters.

**Decision Boundary.** Figure 5 shows the decision boundary predicted with Adam and SAGE on the Spiral dataset. Specifically, we train a multi-layer perceptron with 3 hidden layers, each with a hidden dimension of 100. The decision boundary predicted with SAGE is smoother and has a larger margin than with Adam, suggesting SAGE produces a better generalized model.

## 5.3 COMBINE WITH STATE-OF-THE-ART METHODS

We further show that SAGE is complementary to existing state-of-the-art regularization methods. Specifically, we apply SAGE to SMART (Jiang et al., 2019), a state-of-the-art smoothness-inducing adversarial regularization method. As shown in Table 6, SAGE can further improve upon SMART, suggesting the two techniques are complementary.

| Model | Optimizer | RTE Acc | MRPC Acc/F1 | CoLA Mcc | SST-2 Acc | STS-B P/S Corr | QNLI Acc | QQP Acc/F1 | MNLI-m/mm Acc | Average Score |
|---|---|---|---|---|---|---|---|---|---|---|
| BERT$_{BASE}$ | Adamax | 69.2 | 86.2/90.4 | 57.8 | 92.9 | 89.7/89.2 | 91.2 | 90.9/88.0 | 84.5/84.4 | 82.8 |
| SMART$_{BASE}$ | Adamax | 72.5 | 87.7/91.4 | 59.5 | 93.5 | 90.0/89.6 | 91.9 | 91.7/88.9 | 85.2/85.7 | 84.1 |
| | Adamax-SAGE | **75.1** | **89.0/92.8** | **60.8** | **94.3** | **90.1/89.7** | **92.2** | **91.9/89.1** | **85.9/86.0** | **85.0** |

Table 6: Single task fine-tuning dev results on GLUE.

## 6 DISCUSSION

**SAGE is complementary to Adaptive Gradient Methods.** Our proposed method and the mainstream adaptive gradient methods (e.g., Adam and AdaGrad) are for fundamentally different purposes. The mainstream adaptive gradient methods aim to improve optimization by adapting to the optimization landscape, while SAGE aims to improve generalization by eliminating the weight redundancy. The quantities of our interest (i.e., Eq. (2) and Eq. (3)) are related to the weight redundancy. They are not directly related to the moduli of the objective function, e.g., smoothness, curvature (which are of the interests for optimization). As shown in our experiments (See Section 4), we do not observe any conflicts between the two methods, as SAGE improves the model generalization performance when being combined with several adaptive gradient methods (e.g., Adam).

**Redundant Weights vs. Insufficiently Trained Weights.** Lottery Ticket Hypothesis (Frankle & Carbin, 2018) suggests that, in a randomly initialized network, there exists a well-initialized subnetwork, which outperforms any other subnetworks and matches the full model's performance. This suggests the rest parameters contribute marginally to the model performance. Although the initialization of these parameters may not be satisfactory, SAGE provides them sufficient training so that they can learn to contribute.

## 7 CONCLUSION

We begin with a hypothesis that the redundant parameters can become useful if they are sufficiently trained by desirable optimization strategies. We verify this hypothesis by proposing an adaptive learning schedule – SAGE, which excites the under-fitted parameters to reduce redundancy and regularize the well-fitted parameters to prevent overfitting. We demonstrate that SAGE can benefit model generalization in a wide range of tasks and strengthen various types of optimizers.

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

# A  APPENDIX

## A.1  NATURAL LANGUAGE UNDERSTANDING

### A.1.1  DATA

GLUE is a collection of nine NLU tasks. The benchmark includes question answering (Rajpurkar et al., 2016), linguistic acceptability (CoLA, Warstadt et al. 2019), sentiment analysis (SST, Socher et al. 2013), text similarity (STS-B, Cer et al. 2017), paraphrase detection (MRPC, Dolan & Brockett 2005), and natural language inference (RTE & MNLI, Dagan et al. 2006; Bar-Haim et al. 2006; Giampiccolo et al. 2007; Bentivogli et al. 2009; Williams et al. 2018) tasks. Details of the GLUE benchmark, including tasks, statistics, and evaluation metrics, are summarized in Table 13.

All the texts were tokenized using wordpieces, and were chopped to spans no longer than 512 tokens.

### A.1.2  TRAINING DETAILS

To fine-tune BERT-base and RoBERTa-large models on individual tasks, we append a task-specific fully-connected classification layer to them as in Devlin et al. (2018).

Table 7 present the hyper-parameter configurations. We tune this set of hyper-parameters on a single seed, and report the averaged results obtained with the same configuration over all seeds. For SAGE experiments, We slightly tune $\beta_0$ within a range of $0.1$ on different seeds. We apply a linear weight decay rate of $0.01$ and a gradient norm clipping threshold of $1$ for all experiments. All experiments are conducted on Nvidia V100 GPUs.

| Hyper-param | Experiment | RTE | MRPC | CoLA | SST-2 | STS-B | QNLI | QQP | MNLI |
|---|---|---|---|---|---|---|---|---|---|
| Learning Rate | BERT$_{BASE}$, Adam | 1e-5 | 1e-5 | 1e-5 | 1e-5 | 1e-5 | 1e-5 | 2e-5 | 2e-5 |
| | BERT$_{BASE}$, Adam-SAGE | 1e-4 | 8e-5 | 8e-5 | 3e-5 | 1e-4 | 8e-5 | 4e-5 | 5e-5 |
| | BERT$_{BASE}$, Adamax | 1e-4 | 1e-4 | 1e-4 | 5e-5 | 1e-4 | 1e-4 | 1e-4 | 8e-5 |
| | BERT$_{BASE}$, Adamax-SAGE | 3e-4 | 3e-4 | 2e-4 | 2e-4 | 5e-4 | 5e-4 | 3e-4 | 2e-4 |
| | RoBERTa$_{LARGE}$, Adamax | 5e-5 | 5e-5 | 3e-5 | 1e-5 | 5e-5 | 1e-5 | 1e-4 | 1e-5 |
| | RoBERTa$_{LARGE}$, Adamax-SAGE | 6e-5 | 2e-4 | 8e-5 | 2e-5 | 8e-5 | 3e-5 | 2e-4 | 8e-5 |
| $\beta_0$ | BERT$_{BASE}$, Adam-SAGE | 0.60 | 0.80 | 0.70 | 0.80 | 0.60 | 0.70 | 0.75 | 0.70 |
| | BERT$_{BASE}$, Adamax-SAGE | 0.65 | 0.80 | 0.75 | 0.70 | 0.75 | 0.70 | 0.75 | 0.85 |
| | RoBERTa$_{LARGE}$, Adamax-SAGE | 0.75 | 0.65 | 0.70 | 0.75 | 0.80 | 0.80 | 0.65 | 0.60 |
| Batch Size | BERT$_{BASE}$ | 16 | 8 | 32 | 32 | 32 | 32 | 32 | 32 |
| | RoBERTa$_{LARGE}$ | 16 | 8 | 32 | 32 | 32 | 32 | 32 | 32 |
| Epoch | BERT$_{BASE}$ | 6 | 6 | 6 | 6 | 6 | 3 | 6 | 3 |
| | RoBERTa$_{LARGE}$ | 15 | 6 | 6 | 6 | 10 | 10 | 15 | 3 |
| Dropout | BERT$_{BASE}$ | 0.1 | 0.1 | 0.1 | 0.1 | 0.1 | 0.1 | 0.0 | 0.3 |
| | RoBERTa$_{LARGE}$ | 0.1 | 0.1 | 0.1 | 0.1 | 0.1 | 0.1 | 0.0 | 0.3 |
| Warmup | BERT$_{BASE}$ | 0.1 | 0.1 | 0.1 | 0.1 | 0.1 | 0.1 | 0.0 | 0.1 |
| | RoBERTa$_{LARGE}$ | 0.1 | 0.1 | 0.1 | 0.1 | 0.1 | 0.1 | 0.1 | 0.1 |

Table 7: Hyper-parameter configurations for GLUE experiments. "Epoch" refers to the total training epochs; we adopt early-stopping strategy in practice. "Dropout" refers to classification layer dropout ratio. "Warmup" refers to the ratio of learning rate linear warmup iterations to total training iterations.

### A.1.3  EVALUATION RESULTS

**Statistics of the dev set results.** Table 8 shows the standard deviation of the dev set results.

**Average score computation formula.** For dev set results, we first obtain a score for each task by averaging the scores of all metrics (e.g., Acc and F1) and test sets (e.g., MNLI-m and MNLI-mm) within this task, then compute a task-average score. For test set results, we directly averages scores of all reported metrics following Devlin et al. (2018).

| Model | Optimizer | RTE | MRPC | CoLA | SST-2 | STS-B | QNLI | QQP | MNLI |
|---|---|---|---|---|---|---|---|---|---|
| BERT$_{\text{BASE}}$ | Adam-SAGE | 0.35 | 0.32 | 0.85 | 0.25 | 0.12 | 0.06 | 0.05 | 0.06 |
| | Adamax-SAGE | 0.56 | 0.69 | 0.12 | 0.23 | 0.03 | 0.06 | 0.08 | 0.10 |
| RoBERTa$_{\text{LARGE}}$ | Adamax-SAGE | 0.51 | 0.78 | 0.50 | 0.19 | 0.08 | 0.00 | 0.05 | 0.05 |

Table 8: Standard deviation of the dev set results.

## A.2 NEURAL MACHINE TRANSLATION

### A.2.1 DATA

Table 9 shows the number of sentence pairs in each dataset. We use the standard newstest-2013 and newstest-2014 as dev and test set for WMT'16 En-De. We follow Ott et al. (2019) to split the dev/test sets for IWSLT'14 De-En.

All datasets are encoded using byte-pair encoding (BPE, Sennrich et al. (2016)). We preprocess IWSLT'14 De-En data following *fairseq*[8] and adopt the preprocessed WMT'16 En-De from Google[9].

| Data | Train | Dev | Test |
|---|---|---|---|
| **IWSLT'14 De-En** | 160K | 7283 | 6750 |
| **WMT'16 En-De** | 4.5M | 1061 | 1019 |

Table 9: The number of parallel sentences in NMT datasets.

### A.2.2 TRAINING DETAILS

We adopt the Transformer-base model for both datasets. For IWSLT'14 De-En, we share the decoder and encoder output embeddings. For WMT'16 En-De, we share all the embeddings.

Table 10 presents the hyper-parameter configurations for the best models. We apply a linear weight decay rate of $1 \times 10^{-4}$ and a label smoothing ratio of $0.1$ for all experiments. All experiments are conducted on Nvidia V100 GPUs.

For IWSLT'14 De-En, we report the BLEU score of the best checkpoint using a beam size of $5$ and length penalty of $1$. For WMT'16 En-De, we report the average of the last 10 checkpoints with a beam size of $4$ and length penalty of $0.6$.

| Hyper-param | Experiment | IWSLT'14 De-En | WMT'16 En-De |
|---|---|---|---|
| Learning Rate | Adam
Adam-SAGE | 5e-4
1e-3 | 7e-4
2e-3 |
| $\beta_0$ | Adam-SAGE | 0.8 | 0.4 |
| Batch size | Both | 4096 | 32768 |
| Epoch | Both | 60 | 40 |
| Dropout | Both | 0.3 | 0.1 |
| Warmup | Both | 8000 | 4000 |

Table 10: Hyper-parameter configurations for NMT experiments. "Warmup" refers to the learning rate linear warmup iterations.

---

[8]https://github.com/pytorch/fairseq/blob/master/examples/translation
[9]https://pytorchnlp.readthedocs.io/en/latest/_modules/torchnlp/datasets/wmt.html

## A.3 IMAGE CLASSIFICATION

### A.3.1 DATA

For CIFAR100, we apply random cropping and random horizontal flipping to the training data.

### A.3.2 TRAINING DETAILS

Table 11 present the hyper-parameter configurations for the best models. All experiments are conducted on Nvidia V100 GPUs.

| Hyper-param | Experiment | CIFAR100 | ImageNet |
|---|---|---|---|
| Learning Rate | ViT-B/32, SGD-SAGE | 0.02 | 0.05 |
| | ViT-L/32, SGD-SAGE | 0.02 | 0.08 |
| $\beta_0$ | ViT-B/32, SGD-SAGE | 0.95 | 0.95 |
| | ViT-L/32, SGD-SAGE | 0.85 | 0.95 |
| Training Steps | All | 10000 | 20000 |
| Dropout | All | 0.0 | 0.0 |

Table 11: Hyper-parameter configurations for ViT experiments on CIFAR100 and ImageNet.

## A.4 SUPPLEMENTS FOR METHOD AND ANALYSIS

### A.4.1 ADAM-SAGE ALGORITHM

---

**Algorithm 2** Adam-SAGE ($\odot$ denotes Hadamard product and $\oslash$ denotes Hadamard division)

---

**Input:** Model parameters $\Theta \in \mathbb{R}^J$; Data $\mathcal{D}$; Learning rate schedule $\eta(\cdot)$; Total training iteration $T$;
   Moving average coefficient $\beta_0, \beta_1, \beta_2$.
1: Initialize $\widehat{I}^{(0)}, m^{(0)}, v^{(0)} = \mathbf{0} \in \mathbb{R}^J$.
2: **for** $t = 1, ..., T$ **do**
3:     Sample a minibatch $b^{(t)}$ from $\mathcal{D}$.
4:     Compute gradient $g^{(t)} = \nabla_{\Theta^{(t)}} L(b^{(t)}, \Theta^{(t)})$.
5:     Compute sensitivity $I^{(t)} = |\Theta^{(t)} \odot g^{(t)}|$.
6:     $m^{(t)} = \beta_1 m^{(t-1)} + (1 - \beta_1) g^{(t)}$
7:     $v^{(t)} = \beta_2 v^{(t-1)} + (1 - \beta_2)(g^{(t)})^2$
8:     $\widehat{I}^{(t)} = \beta_0 \widehat{I}^{(t-1)} + (1 - \beta_0) I^{(t)}$.
9:     $\widehat{m}^{(t)} = m^{(t)}/(1 - \beta_1)$
10:    $\widehat{v}^{(t)} = v^{(t)}/(1 - \beta_2)$
11:    $\widehat{I}^{(t)} = \widehat{I}^{(t)}/(1 - \beta_0)$
12:    $U^{(t)} = |I^{(t)} - \widehat{I}^{(t)}|$.
13:    Update $\Theta^{(t+1)} = \Theta^{(t)} - \eta^{(t)}((U^{(t)} + \epsilon) \odot \widehat{m}^{(t)}) \oslash ((\widehat{I}^{(t)} + \epsilon) \odot (\sqrt{\widehat{v}^{(t)}} + \epsilon)) \odot g^{(t)}$.
14: **end for**

---

### A.4.2 IMPLEMENTATION DETAILS FOR SECTION 5.1

Figure 2 experiments: Due to the extremely large model size, we only sample 110K parameters per layer (in total $12 \times 110$K parameters) to calculate the distribution. We select the hyper-parameters that yield the best generalization performance on the BERT-base model, and we evaluate the sensitivity of each parameter using the entire training set.

Figure 4 experiments: Following previous experiment's practice, we randomly sample 110K parameters per layer (in total $12 \times 110$K parameters), and for visualization purposes, we plot 60 randomly selected iterations. We adopt the learning rate corresponding to the best training performance for both SAGE and the baselines.

### A.4.3 IMPLEMENTATION DETAILS FOR SECTION 5.2

Plotting the parameter sensitivity distribution throughout training can be computational expensive. The distribution varies significantly throughout training and often fails to provide a meaningful visualisation. As a result, we compute the structured sensitivity score instead of the parameter sensitivity score. Specifically, we compute a single sensitivity score for each Transformer weight block $\Theta$ at iteration $t$ using the structured counterpart of the parameter sensitivity metric widely adopted in the existing structured pruning literature (Michel et al., 2019; Liang et al., 2021). Following common structured pruning practice, we split Transformer models into 12 feed-forward weight modules and 12 multi-head attention weight modules, and plot the average and variance of the sensitivity of these modules' sensitivity scores throughout the training.

We present the results obtained with the hyper-parameters that yield the best generalization performance on the BERT-base model for both Adamax (Baseline) and Adamax-SAGE (SAGE).

### A.4.4 ABLATION STUDY

To further interpret the role of the parameter sensitivity $I$ and the local temporal variation $U$, we conduct an ablation study on these two factors. Specifically, we check five variants of Eq. (4):

$$\text{Variant 1.} \quad \eta_j^{(t)} = \eta^{(t)}(\widehat{I}_j^{(t)} + \epsilon)(U_j^{(t)} + \epsilon)$$

$$\text{Variant 2.} \quad \eta_j^{(t)} = \eta^{(t)}(\widehat{I}_j^{(t)} + \epsilon)/(U_j^{(t)} + \epsilon)$$

$$\text{Variant 3.} \quad \eta_j^{(t)} = \eta^{(t)}(\widehat{I}_j^{(t)} + \epsilon)$$

$$\text{Variant 4.} \quad \eta_j^{(t)} = \eta^{(t)}/(\widehat{I}_j^{(t)} + \epsilon)$$

$$\text{Variant 5.} \quad \eta_j^{(t)} = \eta^{(t)}(U_j^{(t)} + \epsilon)$$

For Variants 1,2 and 3, we aim to check the performance of giving a high/low-sensitive parameter a high/low, instead of low/high learning rate. Specifically, we place $(\widehat{I}_j^{(t)} + \epsilon)$ in the numerator, so that the learning rates increase for the high sensitive parameters and decrease for low sensitive parameters.

For Variants 4 and 5, we aim to check the performance of eliminating the influence of one of these factors. Specifically, we fix the local temporal variation term to 1 in Variant 4 and fix the sensitivity term to 1 in Variant 5.

### A.4.5 HYPER-PARAMETER STUDY

We investigate the influence of hyper-parameters learning rate and $\beta_0$ on the performance of SAGE (Figure 8). As can be seen, SAGE requires a larger learning rate than the baselines to offset the small scale of the modulation term (the optimal baseline learning rate lies in $5 \times 10^{-5} \sim 1 \times 10^{-4}$ for MNLI, $5 \times 10^{-4} \sim 7 \times 10^{-4}$ for IWSLT 14 De-En and $0.1 \sim 0.2$ for CIFAR10). Furthermore, switching to a larger learning rate requires a lower $\beta_0$ to maintain the same level of performance.

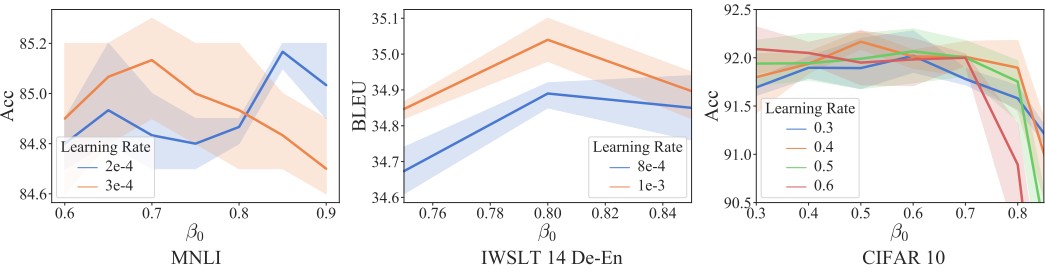

Figure 8: Parameter study on learning rate and $\beta_0$.

All five variants show no clear gain upon the baseline on both RTE and SST-2 datasets after careful hyper-parameter tuning. Specifically, we observe that the Variants 1 and 3 converge very fast at the early stage of training, and then quickly start overfitting. In Variants 2 and 4, the training collapses due to gradient explosion or vanishing.

| Variant Name | Learning Rate Modulating Term | RTE | SST-2 |
|---|---|---|---|
| Adam | $1$ | 63.5 | 92.9 |
| Adam-SAGE | $(U_j^{(t)} + \epsilon)/(\widehat{I}_j^{(t)} + \epsilon)$ | 73.3 | 93.5 |
| Variant 1. | $(\widehat{I}_j^{(t)} + \epsilon)(U_j^{(t)} + \epsilon)$ | 63.5 | 91.2 |
| Variant 2. | $(\widehat{I}_j^{(t)} + \epsilon)/(U_j^{(t)} + \epsilon)$ | Unconverged | Unconverged |
| Variant 3. | $\widehat{I}_j^{(t)} + \epsilon$ | 63.8 | 91.1 |
| Variant 4. | $1/(\widehat{I}_j^{(t)} + \epsilon)$ | Unconverged | Unconverged |
| Variant 5. | $U_j^{(t)} + \epsilon$ | 63.8 | 91.1 |

Table 12: Ablation study on parameter sensitivity and local temporal variations.

| Corpus | Task | #Train | #Dev | #Test | #Label | Metrics |
|---|---|---|---|---|---|---|
| Single-Sentence Classification (GLUE) | | | | | | |
| CoLA | Acceptability | 8.5k | 1k | 1k | 2 | Matthews corr |
| SST | Sentiment | 67k | 872 | 1.8k | 2 | Accuracy |
| Pairwise Text Classification (GLUE) | | | | | | |
| MNLI | NLI | 393k | 20k | 20k | 3 | Accuracy |
| RTE | NLI | 2.5k | 276 | 3k | 2 | Accuracy |
| QQP | Paraphrase | 364k | 40k | 391k | 2 | Accuracy/F1 |
| MRPC | Paraphrase | 3.7k | 408 | 1.7k | 2 | Accuracy/F1 |
| QNLI | QA/NLI | 108k | 5.7k | 5.7k | 2 | Accuracy |
| Text Similarity (GLUE) | | | | | | |
| STS-B | Similarity | 7k | 1.5k | 1.4k | 1 | Pearson/Spearman corr |

Table 13: Summary of the GLUE benchmark.

