# OpenReview forum: "No Parameters Left Behind: Sensitivity Guided Adaptive Learning Rate for Training Large Transformer Models"
_ICLR.cc/2022/Conference — ICLR 2022 Poster_

### Official Review · Reviewer_sSHP · 2021-10-30

**Correctness:** 4
**Technical Novelty And Significance:** 3
**Empirical Novelty And Significance:** 2
**Recommendation:** 6
**Confidence:** 3

**Main Review:**

Strengths:

- **Writing:** The paper is well written and easy to follow.
- **Tasks:** The paper evaluates its method with different optimisers and on different tasks in NLP and image classification.
- **Results:**  The results are quite good; the proposed method surpasses the baseline each time.

Weakness:

- **Architecture:**  The method is evaluated only with transformer architectures. In the context of image classification it would be interesting to evaluate it with CNN and others architecture than ViT.

- **Optimisation:** In image classification, the pre-training procedure used is quite sub-optimal since the paper of Dosovitsky et al. [1] many improvements have been proposed such as the DeiT approach [2]. It would be interesting to see if the proposed method still works when the model is trained with more regularisation and data-augmentation which may also lead to a better use of weights.

- **Image Classification:** Having only results of models pre-trained on ImageNet-21k and fine-tuned with the proposed method on downstream tasks is not very usual in image classification. It would be interesting to have results on ImageNet only where the SAGE method is used during the training.

[1] Dosovitskiy et al, An Image is Worth 16x16 Words: Transformers for Image Recognition at Scale, ICLR 2021

[2] Touvron et al, Training data-efficient image transformers & distillation through attention, ICML 2021

**Summary Of The Paper:**

The paper proposes the SAGE method. The idea is that neural networks have redundant parameters. Some approaches will prune these parameters which has the effect of not decreasing the performance but to decrease the number of parameters. The paper studies if it is possible to learn better these parameters in order to make them more useful for the network. The paper proposes a method that will allow to train differently the parameters of a network in order to have a better use of the weights. The paper is evaluated in NLP and image classification with transformers.


**Summary Of The Review:**

The idea of the paper is interesting and the proposed method seems effective. Nevertheless, more complete experiments in image classification would allow to better evaluate the interest of the proposed method.

---

> ### Author Response · Authors · 2021-11-23
> **Response to Reviewer sSHP**
>
> We thank the reviewer for the detailed comments and questions! Below are our responses:
>
> **The method is evaluated only with transformer architectures. In the context of image classification it would be interesting to evaluate it with CNN and others architecture than ViT. Having only results of models pre-trained on ImageNet-21k and fine-tuned with the proposed method on downstream tasks is not very usual in image classification. It would be interesting to have results on ImageNet only where the SAGE method is used during the training.**
>
> We have explicitly mentioned in our title and Section 1 that we consider Transformer architecture due to its large redundancy. Based on the reviewer’s suggestion, we have also conducted preliminary experiments on training ResNet-20 on CIFAR10/100 from scratch. Below are some experiment results:
>
> | | CIFAR-10 | CIFAR-100 |
> |---|---|---|
> | SGD | 91.56 | 68.31 |
> | SGD-SAGE | 92.41 | 69.06 |
>
>
> **In image classification, the pre-training procedure used is quite sub-optimal since the paper of Dosovitsky et al. many improvements have been proposed such as the DeiT approach. It would be interesting to see if the proposed method still works when the model is trained with more regularisation and data-augmentation which may also lead to a better use of weights.**
>
> Since we only aim to demonstrate that SAGE can be effective in image domain, we choose the most basic training setting. It is definitely the next step to check SAGE’s performance on vision models under more regularisation and data-augmentation techniques.
>
> Worth mentioning, in Section 5.3, we present the performance of combining SAGE with SMART[1], a state-of-the-art adversarial regularization method for language models. The experiment results show that SAGE can further improve upon SMART, suggesting it is complementary to more complicated regularization techniques.
>
> [1] SMART: Robust and Efficient Fine-Tuning for Pre-trained Natural Language Models through Principled Regularized Optimization

---

> > ### Comment · Reviewer_sSHP · 2021-11-29
> > **Thanks for your answer**
> >
> > Thanks for  your answer. The rebuttal addresses some of my concerns. Nevertheless, I have some concerns about the significance of the experimental results.
> > For the ResNet-20 experiments on CIFAR the std's are not reported. The statistics of Figure 7 provided in the rebuttal do not show a significant advantage of Adam over Adam-SAGE given the mean and std of the experiments.
> > However, the idea interesting, well motivated and some of the results reported in the paper seem large enough to be significant.
> > So I'll keep my original score.

---

### Official Review · Reviewer_4pzE · 2021-11-02

**Correctness:** 3
**Technical Novelty And Significance:** 3
**Empirical Novelty And Significance:** 3
**Recommendation:** 8
**Confidence:** 4

**Main Review:**

Strengths:

- The idea is novel, straightforward, well motivated, and easy to implement. Computational cost is low.
- Potential for large impact on model training best practice.
- The experiments are very thorough, demonstrating gains across different settings, and showing that the technique achieves its goal.
- The paper is extremely clearly and carefully written.

Weaknesses:

- The scaling formula seems somewhat ad hoc and hard to characterize. In particular, if the sensitivity of some parameter spikes on a given iteration, it will get a larger update than another parameter with the same moving-average sensitivity that has not spiked.
- There are some hints that the gains might be partly related to regularization (eg, better results on IWSLT than WMT). It would have been good to test the combination with dropout or something similar.
- The hyper-parameters for SAGE are exhaustively tuned, more so than for the baseline adaptive optimizers, so there’s a potential for bias. Figure 7 counters this possibility, but only by showing heatmaps, so it's hard to gauge the actual numbers.

Details:

- Table 3 should show results for your implementation of Adamax that’s comparable to Adamax-SAGE, in addition to the Devlin et al numbers.
- Figure 2: Consider flipping these plots to the standard convention, and stacking them.
- Figure 4: Why do we care that local temporal variation is lower in SAGE than the baselines?

**Summary Of The Paper:**

This paper proposes a method for scaling the learning rate during training in order to encourage all parameters in a neural net to be fully used. Specifically, the updates for parameters are scaled in inverse proportion to how much they affect the loss (their sensitivity); the scaling factor also depends on how stable the estimate of sensitivity is, so that high-sensitivity parameters whose role changes rapidly aren't down-weighted as much. This technique is shown to improve the performance of Transformer models on several different problems, even when used in combination with other adaptive learning rate methods (Adam, Adamax). Analysis experiments verify that the method has the intended effect: compared to the baseline, more parameters have higher sensitivity, and pruning is less effective.


**Summary Of The Review:**

A simple adaptive learning-rate formula that seems to work really well, even on top of other optimizers, as demonstrated by very thorough experiments. Downsides are that the formula isn't particularly well justified, and there are a few potential experimental weaknesses.

---

> ### Author Response · Authors · 2021-11-23
> **Response to Reviewer 4pzE**
>
> We thank the reviewer for the helpful comments! Below are our responses:
>
> **The scaling formula seems somewhat ad hoc and hard to characterize. In particular, if the sensitivity of some parameter spikes on a given iteration, it will get a larger update than another parameter with the same moving-average sensitivity that has not spiked.**
>
> We have explained the reasons for the design of such a formula in Sections 1 and 3. Specifically, a large local temporal variation (“spike”) implies that there exists high uncertainty in the sensitivity. Therefore, the current sensitivity score is not yet a reliable indicator of redundancy. Accordingly, we should avoid decreasing its learning rate no matter if the current sensitivity is large or small.
>
> To better characterize the proposed formula during training, we empirically analyze I (Figure 2, Figure 6), U (Figure 4), learning rate, training loss, and validation loss (Figure 6).
>
> Furthermore, we have tried multiple variants of the proposed formula. We present the performance of all different variants in Appendix A.4.4.
>
> **There are some hints that the gains might be partly related to regularization (eg, better results on IWSLT than WMT). It would have been good to test the combination with dropout or something similar.**
>
> We agree with the reviewer that the gains are partially due to regularization. As stated in Section 1, one of our major motivations is to regularize the well-fitted parameters to prevent overfitting. Accordingly, we design SAGE such that it tends to decrease the learning rate for those well-fitted parameters.
>
> All our experiments are conducted with dropout regularization. Furthermore, in Section 5.3, we combine SAGE with SMART[1], a state-of-the-art adversarial regularization method. The experiment results suggest that SAGE and these regularization methods are complementary.
>
> [1] SMART: Robust and Efficient Fine-Tuning for Pre-trained Natural Language Models through Principled Regularized Optimization
>
> **The hyper-parameters for SAGE are exhaustively tuned, more so than for the baseline adaptive optimizers, so there’s a potential for bias. Figure 7 counters this possibility, but only by showing heatmaps, so it's hard to gauge the actual numbers.**
>
> The statistics of the heatmap are listed as follows:
>
> | Method | Min | Max | 25% Quantile | Mean | 75% Quantile | Median | Std |
> | --- | --- | --- | --- | --- | --- | --- | --- |
> | Adam | 52.71 | 72.89 | 56.68 | 63.64 | 64.98 | 69.31 | 6.97 |
> | Adam-SAGE, beta=0.8 | 52.71 | 73.46 | 63.54 | 64.94 | 65.70 | 69.31 | 6.12 |
> | Adam-SAGE, beta=0.7 | 52.71 | 73.59 | 63.73 | 65.64 | 65.70 | 70.04 | 5.63 |
> | Adam-SAGE, beta=0.6 | 52.71 | 73.59 | 63.54 | 65.89 | 66.79 | 70.04 | 5.50 |
>
> **Table 3 should show results for your implementation of Adamax that are comparable to Adamax-SAGE, in addition to the Devlin et al numbers.**
>
> The test results for Adamax are listed as follows:
>
> | Method | MNLI-m/mm | QQP | QNLI | SST-2 | RTE | CoLA | MRPC | STS-B | Avg |
> | --- | --- | --- | --- | --- | --- | --- | --- | --- | --- |
> | Adamax | 84.7/83.6 | 71.1 | 90.6 | 93.4 | 66.8 | 54.0 | 88.6 | 86.6 | 79.9 |
>
> We have updated the paper accordingly.
>
> **Figure 4: Why do we care that local temporal variation is lower in SAGE than the baselines?**
>
> This comparison intends to demonstrate that adjusting the learning rate based on the local temporal variation (U) is very important to the success of SAGE. Specifically, Figure 4 shows that by adjusting the training based on U, U decreases gradually. This means that the sensitivity score (I) stabilizes and becomes a reliable indicator of redundancy. Based on a more reliable I, the learning rate can be more accurately adjusted. In contrast, without adjusting the training based on U, I constantly has a large U, and is less reliable.
>
> **Figure 2: Consider flipping these plots to the standard convention, and stacking them.**
>
> Does the reviewer mean to overlay the two distributions? The reviewer may clarify and we can make changes in the next version.

---

### Official Review · Reviewer_MzBV · 2021-11-02

**Correctness:** 3
**Technical Novelty And Significance:** 4
**Empirical Novelty And Significance:** 4
**Recommendation:** 6
**Confidence:** 4

**Main Review:**

I think that overall the paper is good as it raises and studies an important point: redundancy in parameters is not an axiom we have to accept.
However, after this introduction and motivation, it reads more like a typical optimizer paper introducing a new optimizer based on hand-wavy intuitions. This is unfortunate and not rigorous.

What follows are detailed comments:
1) The use of Taylor approximation is only good close to the operating point for nonlinear functions. However Theta_j,-j may be very large, and thus the approximation may be very bad.
2) I do not agree that the memory and computational costs of SAGE are "marginal". The EMA "I-hat" is a full copy of the model since we need one EMA per parameter, especially for large models this is substantial overhead. I would further appreciate timings to verify that indeed the computational overhead is "marginal".
3) It is not clear to me in what scale the quantities U and I-hat are, and hence, the multiplier to the learning-rate. Because of this, it is not clear how one needs to change the original learning-rate "eta" when one turns on SAGE.
4) I am confused by Fig5, it does not look like the two-moon dataset at all to me? Please image-search "two moon dataset" and compare.

minor nitpicks:
5) Fig1: Percent should be [0,100].
6) I do not agree that one can conclude from the results that SAGE is more effective for small datasets than for large ones, since the "points" betwen the datasets/tasks do not live on the same scale.
7) All experimental evaluation is about fine-tuning of already pre-trained transformer models. This should be more accurately reflected in the title, for example by replacing the word "training" by "fine-tuning". It would further be interesting to see how well it works when training from scratch; even if it does not work, it is still a valuable fine-tuning method, and stating this may save a lot of people a lot of resources and time.
8) In Figure 6, since we have per-parameter learning-rates, which one is shown for SAGE?
9) Typo: "smoothier" -> "smoother"

**Summary Of The Paper:**

The paper argues that redundant or "useless" parameters are not an axiom we should take for granted, but rather a symptom of current optimization settings. The authors propose an adaptive learning-rate schedule that specifically aims to eliminate redundant parameters. Through extensive experiments on fine-tuning transformers, they show that indeed this method (SAGE) does reduce redundancy and also slightly improves results.

**Summary Of The Review:**

For an "optimizer" paper, it is weak: hand-wavy motivation, no real derivation of update rule or even convergence proofs, and experiments only in one very specific domain: transfer of pre-trained transformer models. I am not convinced that it will be generally useful at all.

However, the paper raises a very important point: we should not take redundant parameters as a necessity, and it does propose a method to avoid them. I find this point important enough to still suggest acceptance.

---

> ### Author Response · Authors · 2021-11-23
> **Response to Reviewer MzBV**
>
> We thank the reviewer for the insightful comments! Below are our responses:
>
> **For an "optimizer" paper, it is weak: hand-wavy motivation, no real derivation of update rule or even convergence proofs.**
>
> Our paper is not an “optimizer” paper. The motivation of SAGE is to improve model generalization performance by eliminating the weight redundancy. The quantities of our interest are not directly related to the moduli of the objective function, e.g., smoothness, curvature (which are of interest for optimization). Furthermore, we empirically verified that SAGE indeed leads to more sufficient training (Section 5.1) and thus better generalization performance (Section 5.2).
>
> We agree with the reviewer on the importance of the derivation of the update rule and the convergence guarantee. We have tried multiple variants of update rules and present the performance of all variants in Appendix A.4.4. We will also look into the theoretical analysis as our next step. However, the convergence guarantees of the “optimizer” papers are usually established upon Lipschitz continuous gradient and bounded stochastic gradient assumptions, which do not hold for Transformer models in practice. Such convergence to the first-order stationary solutions also does not imply any theoretical guarantees on the model generalization performance.
>
> **Comment 1**
>
> The importance score based on the first-order Taylor expansion is introduced in [1], and has been shown to well-approximate the sensitivity in various pruning approaches [2-4]. As $\Theta_{j, -j}$ is generally small in scale (e.g., the average absolute value is 0.0325 when fine-tuning BERT-base on MNLI), dropping high order terms can still provide a sufficiently accurate approximation.
>
> Within the same model, the larger-scaled weights generally have larger sensitivity. These weights can tolerate relatively large approximation errors, and their importance scores are unlikely to be small.
>
> [1] Molchanov, Pavlo, et al. "Pruning convolutional neural networks for resource efficient inference."
>
> [2] Ding, Xiaohan, et al. "Global sparse momentum sgd for pruning very deep neural networks."
>
> [3] Molchanov, Pavlo, et al. "Importance estimation for neural network pruning."
>
> [4] Michel, Paul, Omer Levy, and Graham Neubig. "Are sixteen heads really better than one?."
>
> **Comment 2**
>
> We agree with the reviewer that the memory overhead is a copy of the model, but it is not substantial compared with the existing memory cost. For example, in Adam, we need to store gradients, first-order momentum, and second-order momentum, each requiring a full copy of the model. Moreover, the extra copy can be partitioned into different GPUs with existing DDP accelerators, e.g., DeepSpeed, making the additional cost marginal.
>
> We list the training time for several tasks as the following, which shows that the computational overhead is around 6%-8%.
>
> | Training time per epoch (second) | MNLI | SST-2 | RTE |
> |---|---|---|---|
> | Adamax, FP32 | 2439.2 | 238.0 | 29.1 |
> | Adamax-SAGE, FP32 | 2606.3 | 258.2 | 30.8 |
>
> **Comment 3**
>
> Figure 4 plots the average scale of U when fine-tuning BERT-base on GLUE tasks. The scale is generally around $3\times 10^{-7}$ at the beginning of the training, then decreases through time.
>
> We further track the scale of $\hat{I}$. It is generally around $1\times 10^{-6}$ at the beginning of the training, then decreases through time.
>
> The $U/\hat{I}$ ratio is usually smaller than 1, and we empirically find that we need to adjust the original learning rate to be several times larger (as reported in Table 7).
>
> **Comment 4**
>
> We apologize for the inaccuracy. We use the “Spiral Dataset” publicly available [1]. It essentially changes the “Two Moon” dataset’s “theta = np.sqrt(np.random.rand(N))*pi” to be “theta = np.sqrt(np.random.rand(N))*2*pi”.
>
> [1] https://gist.github.com/45deg/e731d9e7f478de134def5668324c44c5
>
> **Comment 6**
>
> We agree with the reviewer and believe we have never made such a claim. In Table 1, 2 points of improvement on small datasets are larger than improvements obtained by many existing fine-tuning methods. As a result, we claim that SAGE is very effective on small datasets.
>
> **Comment 7**
>
> The NMT experiments in Section 4.2 evaluate the **trained-from-scratch** transformer models. Table 4 shows SAGE improves around 0.6 and 0.4 points on the IWSLT and the WMT test set, respectively.
>
> We explicitly consider Transformer architecture due to its large redundancy. We have also conducted preliminary experiments on training ResNet-20 on CIFAR10/100 from scratch:
>
> | | CIFAR-10 | CIFAR-100 |
> |---|---|---|
> | SGD | 91.56 | 68.31 |
> | SGD-SAGE | 92.41 | 69.06 |
>
> **Comment 8**
>
> The orange curve is shown for SAGE. Specifically, we plot the average and variance of SAGE’s learning rate over all parameters in the model. The implementation details can be found in Appendix A.4.3.
>
> **Comment 5 and 9**
>
> We have updated the paper accordingly.

---

### Official Review · Reviewer_NSqH · 2021-11-03

**Correctness:** 4
**Technical Novelty And Significance:** 2
**Empirical Novelty And Significance:** 3
**Recommendation:** 6
**Confidence:** 4

**Main Review:**

Strengths
1. The paper is well-written and easy to follow. The method section provides helpful intuitions behind the algorithm design.
2. The idea is interesting since it explores a different direction in dealing with redundant parameters.
3. Experiments use multiple benchmarks, including both language and vision, and results show noticeable performance improvements.
4. SAGE is orthogonal to the existing adaptive gradient methods. Jointly using them can bring more gains.

Weaknesses
1. Table 1 shows that models with different percentages of redundant parameters have similar performance, implying that performance may not be proportional to "well-trained" parameters. It means that making more parameters "well-trained" does not necessarily improve the performance. This seems not totally in line with the paper's motivation.
2. Figure 5 is not straightforward to visualize the difference. The curve is drawn only on the right subfigure.
3. Sufficiently training all parameters seems a double-edged sword. According to Figure 3, models trained with SAGE are susceptible to parameter pruning. In general, we want efficient and compact models in deployment. Will SAGE be harmful to developing efficient deployment models?

**Summary Of The Paper:**

This paper focuses on the parameter redundancy issue in large transformer architectures. Instead of pruning redundant parameters, it strengthens training them to make them contribute better performance. To this end, it proposes an adaptive learning rate algorithm SAGE, which automatically scales the learning rate for each parameter based on its sensitivity. The sensitivity is approximated by the dot product between parameters and their gradients. An exponential moving average is used to track the sensitivity scores to reduce uncertainty in mini-batch training. The algorithm is applied to fine-tuning pre-trained transformer models in benchmarks for natural language understanding (NLU), neural machine translation (NMT), and image classification.

**Summary Of The Review:**

This paper proposes SAGE, an adaptive learning rate schedule to train redundant parameters more sufficiently to improve model generalization. SAGE, together with existing adaptive optimizers, shows effectiveness in a wide range of downstream tasks. The main concern is whether SAGE has adverse effects in getting efficient deployment models.

---

> ### Author Response · Authors · 2021-11-23
> **Response to Reviewer NSqH**
>
> We thank the reviewer for the detailed comments! Below are our responses:
>
> **Table 1 shows that models with different percentages of redundant parameters have similar performance, implying that performance may not be proportional to "well-trained" parameters. It means that making more parameters "well-trained" does not necessarily improve the performance. This seems not totally in line with the paper's motivation.**
>
> We believe that the reviewer has some misunderstandings here. For Table 1, we train multiple models with different learning rates, select the top 30% of most redundant parameters from each model, and compute the **percentage of overlapping** across these redundant parameters. For example, among the 2 models’ top 30% of most redundant parameters, only 59.8% of parameter indices are the same. This suggests that training strategy plays a role in determining which parameter ends up being redundant.
>
> **Figure 5 is not straightforward to visualize the difference. The curve is drawn only on the right subfigure.**
>
> We have updated the paper accordingly.
>
> **Sufficiently training all parameters seems a double-edged sword. According to Figure 3, models trained with SAGE are susceptible to parameter pruning. In general, we want efficient and compact models in deployment. Will SAGE be harmful to developing efficient deployment models?**
>
> We believe that the reviewer brings up a good point. As SAGE improves the model’s overall generalization performance, the lightly compressed model trained with SAGE performs better than without SAGE (Figure 3). However, its performance becomes more susceptible to pruning when the model is compressed heavier.
>
> However, we do not see it as a limitation. It is up to us to determine how to leverage SAGE so that it can help produce a better-generalized compact model. For example, a pruned BERT-large with 50% of remaining weights (180M) achieves an accuracy of 85.3/85.7 on MNLI-m/mm$^1$ , while training a BERT-base (110M) with SAGE+SMART achieves 85.9/86.0 (Table 6). Another straightforward idea is to fine-tune / distill the pruned model using SAGE as the optimizer to further improve its generalization performance.
>
> $^1$ We conduct an one-shot structured pruning based on the importance score in [1].
>
> [1] Michel, Paul, Omer Levy, and Graham Neubig. "Are sixteen heads really better than one?." arXiv preprint arXiv:1905.10650 (2019).

---

> > ### Comment · Reviewer_NSqH · 2021-12-01
> > **Thanks for the reply**
> >
> > Thanks for the reply, which has cleared my questions about Table 1 and Figure 5. But my concern regarding the model pruning and compression remains. According to Figure 3 (the bottom row), the SAGE-trained model is more susceptible to parameter pruning than the baseline model. Besides, the comparison on MNLI-m/mm is only on one dataset and not listed in the paper. I suggest adding some new experiments and analyses on this topic. Therefore, I keep my original rating.

---

### Decision · Program_Chairs · 2022-01-20

**Decision:**

Accept (Poster)

**Comment:**

The paper observes that the number of redundant parameters is a function of the training procedure and proposes a training strategy that encourages all parameters in the model to be trained sufficiently and become useful. The method adaptively adjusts the learning rate for each individual parameter according to its sensitivity (a proxy for the parameter's contribution to the model performance). The approach encourages the use of under fitted parameters while preventing overfitting in the well-fitted ones. Experimental results are presented covering a wide range of tasks and in combination with several optimisers, showing improvements in model generalization.

The paper is very well written and easy to follow (as mentioned by Reviewers NSqH, 4pzE and sSHP).

The authors provided a strong rebuttal including new experiments, like training using CNN based architectures (as requested by Reviewers sSHP and MzBV). Reviewer sSHP requested these results to be reported with STD, the AC encourages the authors to do so for the camera ready.

Reviewer MzBV points out that the paper could be improved by giving a motivation of the update rule and proving convergence. However, still recommends accepting the paper due to the novelty in the idea of not taking redundant parameters as something inevitable and devising an effective strategy to improve it. This idea was also appreciated by the other reviewers. While the AC agrees that adding these points would improve the work, it takes as valid the point made by the authors. Namely, that the intuition behind the update rule is quite clear, and many other reasonable variants were ablated (in Appendix A.4.4). Furthermore, the empirical evidence shows that the method improves generalization.

Reviewer NSqH points out that while SAGE improves the model’s generalization performance for lightly compressed models, its performance becomes more susceptible to pruning when the model is compressed heavier. While the authors responded with good points, the AC encourages them to follow the reviewer’s advice and incorporate further experiments studying this issue (e.g. other datasets).

In sum, the paper proposes a simple and effective method that is able to improve generalization of large scale models. All four reviewers recommend accepting the paper. The AC agrees and encourages the authors to incorporate the requests mentioned above.